# Functional Characterization of the Cystine-Rich-Receptor-like Kinases (*CRKs*) and Their Expression Response to *Sclerotinia sclerotiorum* and Abiotic Stresses in *Brassica napus*

**DOI:** 10.3390/ijms24010511

**Published:** 2022-12-28

**Authors:** Rehman Sarwar, Lei Li, Jiang Yu, Yijie Zhang, Rui Geng, Qingfeng Meng, Keming Zhu, Xiao-Li Tan

**Affiliations:** 1School of Food Science and Biological Engineering, Jiangsu University, Zhenjiang 212013, China; 2School of Life Sciences, Jiangsu University, Zhenjiang 212013, China

**Keywords:** genome-wide, *Sclerotinia sclerotiorum*, abiotic stresses, receptor-like kinases

## Abstract

Cysteine-rich receptor-like kinases (*CRKs*) are transmembrane proteins that bind to the calcium ion to regulate stress-signaling and plant development-related pathways, as indicated by several pieces of evidence. However, the *CRK* gene family hasn’t been inadequately examined in *Brassica napus*. In our study, 27 members of the *CRK* gene family were identified in *Brassica napus*, which are categorized into three phylogenetic groups and display synteny relationship to the *Arabidopsis thaliana* orthologs. All the *CRK* genes contain highly conserved N-terminal PKINASE domain; however, the distribution of motifs and gene structure were variable conserved. The functional divergence analysis between *BnaCRK* groups indicates a shift in evolutionary rate after duplication events, demonstrating that *BnaCRKs* might direct a specific function. RNA-Seq datasets and quantitative real-time PCR (qRT-PCR) exhibit the complex expression profile of the *BnaCRKs* in plant tissues under multiple stresses. Nevertheless, *BnaA06CRK6-1* and *BnaA08CRK8* from group B were perceived to play a predominant role in the *Brassica napus* stress signaling pathway in response to drought, salinity, and *Sclerotinia sclerotiorum* infection. Insights gained from this study improve our knowledge about the *Brassica napus CRK* gene family and provide a basis for enhancing the quality of rapeseed.

## 1. Introduction

Oilseed rape (*Brassica napus*), with profuse vegetable oil, nutrient-rich meal, and a source of biofuel, is considered a highly economically important crop worldwide. However, the yield of this crop suffers dramatically from various biotic and abiotic stresses [1]. Furthermore, fungal pathogens such as *Alternaria brassicae* and *Sclerotinia sclerotiorum* (*S. sclerotiorum*) causing black spots and stem root, potentially affect *B. napus* yield and harvest index [2,3]. To fight against these stresses, plants utilize different defensive mechanisms, including modulation in Ca^2+^ levels, reactive-oxygen-species (ROS), and regulation of stress-induced transcriptional factors (TFs), which mediates the defense-related genes expression pattern in response to pathogens infection [4,5]. It was reported that receptor-like kinases (RLKs) direct defense response and plant development by establishing a signaling network between the plant membrane and the nucleus [6,7]. Previous studies have reported that *Arabidopsis thaliana* (*A. thaliana*) and *Oryza sativa* (*O. sativa*) contain more than 1000 copies of *RLKs* in their genome [8,9,10]. *RLKs* include more than one domain, categorizing this gene family into several distinct subfamilies [8]. For example, in *A. thaliana*, the lysine-containing ectodomain of CHITIN ELICITOR RECEPTOR KINASE 1 (CERK1) directly binds with fungal chitin to initiate a plant immune response [11]. In contrast, the FLAGELLIN SENSITIVE 2 (FLS2), a leucine-rich transmembrane receptor kinase, interacts with bacterial active epitope flagellin (flg22), resulting in the activation of plant defense response against several pathogens [12].

In plants, cysteine-rich receptor-like kinases (*CRKs*) are a major subfamily of *RLKs* entangled in apoptosis and disease resistance. *CRKs* comprise a single-transmembrane domain, an extracellular domain, and a Ser/Thr protein kinase domain [13]. It has been reported that a group of the cystine-rich-receptor-like kinase domain of the *CRKs* encoding the C-8X-C-2X-C motif is induced by multiple stresses, including pathogen infection, oxidative stress, and abiotic stresses [14,15,16]. Furthermore, the N-terminal region of the *CRKs* holds a myristoylation site, and a mutation in this site positions the tomato *CRK* (*LeCRK1*) to the nucleus [17], which suggests the importance of *CRKs* myristoylation site for their localization in the plasma membrane [18]. Characterization and identification of the *CRK* gene family have been evaluated in several plant species, including *Gossypium barbadense* [19], *Solanum lycopersicum* (*S. lycopersicum*) [20], *O. sativa* [21], and *Malus domestica* [22] (Table 1), but only half of the *CRKs* biological function have been elucidated. These reports demonstrated the involvement of a few *CRKs* in hormonal signaling pathways, plant growth, and reaction to several abiotic and biotic stresses [21,23]. For example, overexpression of *CRK4* and *CRK5* in *A. thaliana* suggests a possible role in the leaf development and initiates the pattern-triggered immunity response to pathogen infection [24,25,26], whereas the overexpression of *AtCBK3*, which is also known as *AtCRK1* resulted in enhanced thermotolerance [27]. Additionally, in the previous study, the interaction of *CRK3* with cytosolic glutamine synthetase (*GLN1*) was reported to remobilize nitrogen during leaf senescence [28]. In contrast, the *CRK1* and *CRK5* were found to negatively regulate ABA-signaling to confer drought tolerance [29,30,31]. Furthermore, a *CRK* gene from *A. thaliana* (*AtCRK6, 7*) and *HvCRK1* from *Hordeum vulgare* were reported to promote a ROS-mediated response to powdery mildew [32,33,34], and a *CRK* gene from wheat (*TaCRK1*) showed higher expression in response to *Rhizoctonia cerealis* [35]. Results from these studies revealed the numerous roles of the *CRK* genes in the physiological processes of plant development. However, the functional study of *CRK* in response to *S. sclerotiorum* and abiotic stresses in *B. napus* remains largely unknown.

In the present report, the genome-wide identification of all the members of the *CRK* gene family in *B. napus* was identified and summarized, and the expression profile in response to drought, freezing, salinity, and *S. sclerotiorum* treatments was predicted. The findings gained from this study provide useful insights into the biological functions of *CRK* members in *B. napus*.

## 2. Result

### 2.1. Identification, Phylogenetic Analysis, and Structural Characterization of B. napus CRK Family

To understand the evolutionary mechanism of the *CRK* gene family in *B. napus*, we extensively searched and retrieved information about the eight *A. thaliana CRK* genes from a previous study (Table 1). The peptide sequences of *AtCRK* genes were used to search in the *B. napus* genome browsers GENOSCOPE [42] and BnPIR [43,44]. A total of 27 *CRK* gene family members were predicted in the *B. napus* genome and designated as *BnaCRK1* to *BnaCRK8*, which corresponded to each member of the *A. thaliana* orthologs (Appendix A). In contrast, we also identified 15, 11, 14, and 24 *CRK* gene family members in *Brassica rapa, Brassica oleracea, Brassica nigra,* and *Brassica juncea*, respectively. The amino acid length of the BnaCRK proteins varied from 186 to 622, while the molecular weight (MW) ranged from 20,424 kDa to 69,742 kDa. The isoelectric point (pI) value ranges from 7.97 to 10.16, suggesting that the BnaCRK proteins were highly alkaline (Appendix A). In contrast, the aliphatic index (AI) was between 53.6 to 91.4, which shows the high thermal stability of the BnaCRK proteins. Furthermore, the negative value of the grand average of hydrophobicity (GRAVY) shows that all BnaCRK proteins belong to the hydrophilic group, while the instability index (II) value was above 40, indicating that all BnaCRK proteins were unstable. Furthermore, among the 27 *BnaCRKs*, all members from group B, two from group A, and *BnaA04CRK3* from group C were predicted to contain myristoylation sites (Appendix A).

The full-length peptide sequences of the *CRK* genes from *A. thaliana*, *B. napus*, *B. rapa*, *B. oleracea*, *B. nigra*, and *B. juncea* were aligned to construct a phylogenetic tree. As shown in Figure 1, we found that the 99 *CRK* genes were clustered into three groups (groups A to C). Group A contains the *CRK1*, *CRK5*, and *CRK7* genes. Group B holds the *CRK2*, *CRK4*, *CRK6*, and *CRK8* genes, whereas group C contains the *CRK3* genes. In *B. napus,* 10, 15, and 2 members of the *CRK* gene family were organized into groups A, B, and C, which are closely related to *A. thaliana* orthologs (Appendix A). To examine the structural features, we analyzed the exon and intron composition of the *BnaCRK* gene family (Appendix A). The number of exons in the *BnaCRKs* ranges from 1 to 11, in which most of the *CRK* genes contain ten introns. For instance, all group A and group C members contain 11 exons and 10 introns, except for the *BnaA04CRK7,* which comprises 10 exons and 9 introns. In contrast, members in group B hold 2 to 11 exons and 1 to 10 introns, except for the *BnaCRK6* genes, which contain no intron and only 1 exon. However, the distribution of exons and gene size within a subfamily is varied, which might be due to the insertion of the introns. Overall, the numbers of exons and introns are similar to the *A. thaliana CRK* gene family (Appendix A).

### 2.2. Motif Prediction and Distribution Analysis of BnaCRKs

Alignment of the protein sequence of the *A. thaliana, B. napus, B. oleracea, B. nigra, B. rapa,* and *B. juncea* display the highly conserved N-terminus PKINASE domain (Appendix A). It has been predicted that the binding of the Ca^2+^ enhances kinase activity, and the presence of myristoylation sites in the N-terminus of the BnaCRK proteins is reported to stimulate the interaction of defense-related proteins and membrane compatibility [45,46,47]. To further investigate the structural characteristics of the BnaCRKs, the 20 distinct motifs within 50 to 8 amino acids were predicted (Figure 2). The seq logos of predicted motifs are shown in Appendix A. In our motif prediction analysis, we perceived that the members of the *BnaCRK* gene family consist of 6 to 18 putative conserved motifs. All members contain motifs 8, 12, and 14, while few motifs are predicted in some *BnaCRKs* (Figure 2). For example, motif 16 was only found in *BnaCRK3*, *BnaCRK6*, *BnaA08CRK8*, and *BnaC03CRK8*, while motif 17 was present only in *BnaCRK2* and *BnaCRK8*. Furthermore, motifs 18 and 20 were only found in *BnaCRK4* and *BnaCRK6*, except for *BnaC07CRK4*, which excludes motif 18. However, within the same subfamily, the architecture and motif composition were comparable (Figure 2), implying that *CRK* genes within the same subfamily might have similar biological roles.

### 2.3. Functional Divergence Analysis and Protein 3D Structure Prediction

To examine the functional diversity between the subfamilies of *BnaCRKs*, we conducted Type I (*ϴI*) and Type II (*ϴII*) functional divergence utilizing the protein sequences of the BnaCRKs. As shown in Table 2, we found that the coefficient *ϴI* values of group A/B, group B/C, and group A/C were 0.12, 0.20, and −0.22, respectively, and the LRT values among group A/B and group B/C were 0.9 and 1.2, respectively, which indicates that Type I functional divergence might exist in the *BnaCRK* subfamilies. Moreover, four, two, and five amino acid sites involved in *ϴI* functional divergence were also predicted in group A/B, group A/C, and group B/C, respectively, suggesting that these amino acid sites might direct the alteration in functional limitations, which was associated with the evolutionary constrains after duplication events. Conversely, the *ϴII* functional divergence represents the modulation of amino acid chemical and physical properties after gene duplication events [48]. Our results predicted that the *ϴII* coefficient values between different subfamilies of the *BnaCRKs* were small but higher than 0 (Table 2). Additionally, in *ϴII* functional divergence, group A/B, group A/C, and group B/C contain 29, 21, and 42 critical amino acid sites, respectively, in which the 3I, 100K, 104P, 113R, and 114A amino acid sites were relatively conserved in both *ϴI* and *ϴII*, suggesting that the altered selective constraints between different subfamilies of the *BnaCRKs* might happen at these sites (Table 2).

To further investigate the location of these critical amino acids in BnaCRK proteins, we utilized the BnaA03CRK1 three-dimensional model with the normalized C score >3, which is considered the best template for BnaA03CRK1 three-dimensional structure prediction (Figure 3). As shown in the model, we found that 20 critical amino acids were dispersed on the N-terminal region of the BnaA03CRK1, in which only 6 critical amino acid sites, 116E,124G,130C,132A,138T, and 139G, were present on the PKINASE domain of BnaA03CRK1, indicating the vulnerability of the PKINASE domain to positive selection during the evolution of the *BnaCRK* gene family. However, no important amino acid sites were observed in the C-terminal region of the BnaA03CRK1. Furthermore, 25 α-helicases were found in the putative structure of BnaA03CRK1 protein, in which 7 α-helicases and 8 β-strands were predicted in the N-terminal PKINASE domain (Figure 3B).

### 2.4. Synteny and Duplication Prediction of BnaCRK Gene Family in B. napus

In order to predict the location of the *BnaCRKs*, we plotted the members of the *BnaCRK* gene family onto the genome of *B. napus*. As detailed in Figure 4A, we observed that the 27 members of the *BnaCRK* gene family were distributed on the 14 chromosomes of the *B. napus*. Among them, 13 members were located on the A genome, whereas the other 14 members were localized on the C genome, in which chromosomes C03, C04, and C08 contain three members of the *BnaCRK* gene family. In contrast, chromosomes A01, A03, A04, A05, A06, A09, and C01 hold two members, while A08, C05, C07, and C09 contain only one member (Figure 4A). The evolution of the genome is entirely dependent upon the duplication of the genes, which alleviates the initiation of a new gene family with a new function [49]. Tandem and segmental duplication are considered the major causes of the function and expansion of the gene family during duplication events [50]. To better understand the expansion patterns of the *CRK* gene family, we utilized the MCScanX program to predict the duplication event in the *B. napus* genome. As shown in Figure 4B, we predicted 42 segmental duplication pairs across *B. napus* chromosomes except for scaffoldC02, scaffoldA10, scaffoldA07, scaffold A02, and scaffoldC06. Furthermore, we also identified that the sequence similarity between the all-segmental duplication pairs was highly conserved (>90%). However, no tandem duplication pair of *BnaCRKs* were detected across all *B. napus* chromosomes. Taken together, the results indicate that segmental duplication was the leading cause for the expansion of the *BnaCRK* gene family in *B napus*, and different homologous pairs of genes on *B. napus* chromosomes favor the higher conservation of the *CRK* gene family in *B. napus* (Figure 4).

To further examine the evolutionary mechanism of *BnaCRK* genes, the comparative collinearity map associated with *B. nigra*, *B. oleracea*, *B. rapa,* and *A. thaliana* was created (Figure 5; Appendix A). The 27 members of the *B. napus CRK* gene family were identified to show collinear gene pairs with *A. thaliana* (8), *B. rapa* (15), *B. oleracea* (11), and *B. nigra* (14) members of the *CRK* gene family. For instance, eight *A. thaliana CRK* genes (*AtCRK1, AtCRK2, AtCRK3, AtCRK4, AtCRK5, AtCRK6, AtCRK7,* and *AtCRK8*) corresponded to the five, eight, two, five, four, five, three, and eight orthologous copies of the *BnaCRKs*, respectively (Figure 5). In addition, a total of 68, 70, and 71 collinear gene pairs were predicted among *B. napus/B. oleracea, B. napus/B. rapa,* and *B. napus/B. nigra*, respectively (Appendix A). However, *BnaC03CRK1* was found to have no collinearity relationship with *B. oleracea*, *B. nigra, B. rapa,* and *A. thaliana*, implying that a single orthologous copy of the *CRK* contributed to the evolution and functional diversification of the *CRK* gene family in *B. napus* genome. In addition, the selection pressure of the duplicated genes was also measured by calculating the Ka/Ks ratio. According to the results, we have observed that the Ka/Ks ratio of the *BnaCRKs* was significantly lower than 1, which suggests that the *CRK* gene family in *B. napus* experienced an intense purifying selection (Appendix A). However, some limitations may bring the Ks/Ka ratio to <1 [51,52]. Therefore, different site models were performed in the CODEML program to analyze the selection pressure on a single amino acid codon [53,54] (Appendix A). The one ratio model M0 was selected to predict *ꟺ* values across all amino acid sites. According to the results, we obtained the M0 value *ꟺ* = 0.152, which indicates that the *BnaCRK* gene family experienced a strong purifying selection. Additionally, to assume the dN/dS value difference within the codon sites model, M0 was compared with model M3. As a result, the log-likelihood 2∆InL = −415.087 displays a significant difference (*p* < 0.01) and suggests that the *CRK* gene family in *B. napus* underwent selective pressure across various sites (Appendix A). Furthermore, one and four positive selection sites were identified by model M2 and model M8, respectively, in which site 24V was present in both models. Overall, the *CRK* gene family in *B. napus* underwent a strong purifying selection (Appendix A).

### 2.5. BnaCRK Protein-Protein Interaction Network Prediction

To investigate the role of the BnaCRKs with their interacting targets, we utilized the AtCRKs orthologs to construct a network. As a result, members from group A were found to interact more with seven target proteins than others, in which BnaCRK1 showed interaction with HEAT SHOCK FACTOR 1A (HSF1) and SERINE/THREONINE PHOSPHATASE 7 (PP7), while BnaCRK5 was perceived to be involved in strong interaction with ABA-AND OSMOTIC-STRESS-INDUCIBLE RECEPTOR-LIKE CYTOSOLIC KINASE1 (ARCK1), CYSTEINE-RICH RLK 13 (CRK13), CYSTEINE-RICH RLK 19, CYSTEINE-RICH RLK 20, and CYSTEINE-RICH RLK 36 (Figure 6A). A previous study reported that the interaction of AtCRK1 with PP7 and HSF1 positively regulated the salinity and heat shock responses [27,55], whereas the interaction of AtCRK5 with ARCK1 and CYSTEINE-RICH RLKs might negatively mediate the abscisic acid (ABA) and osmotic stress signal transduction. In contrast, EPSP and CUPULIFORMIS5 (CP5) show powerful interaction with BnaCRK5. However, the interaction response is still unclear. Similarly, GLN1, which encodes a glutamine synthase, shows interaction with BnaCRK3, suggesting the putative role of BnaCRK3 in leaf senescence [28] (Appendix A). Additionally, PROTEIN PHOSPHATASE 2C (PP2C) was also found to interact with all the BnaCRKs, indicating the possible function of the CRKs in the development and transduction signaling pathways [56] (Figure 6B). The outcomes disclosed that the BnaCRPKs are the potential regulator of plant development and are involved in response to various environmental cues.

### 2.6. Expression Profile of the BnaCRKs in Different Organs of B. napus

A few reports have shown that *CRKs* play a fundamental role in different stages of plant development [37,57]. To explore the possible function of *CRKs* in *B. napus* development, we utilize the microarray data from the *B. napus* genome browser BnPIR [43,44] to investigate the expression profile of the *BnaCRK* gene family in organs of the *B. napus* variety Zhongshuang 11 (ZS11). As presented in Appendix A, we observed that *BnaA01CRK2*, *BnaA05CRK2*, *BnaC08CRK5*, and *BnaA09CRK5* showed mild transcriptional levels in all tissues (Appendix A), while the remaining 27 *BnaCRKs* showed expression in root, bud, filament, petal, pollen, sepal, cotyledon, vegetative-rosette, leaf, seed, silique, and stem (Appendix A). In detail, the members from group A (*BnaA03CRK1*, *BnaA01CRK5*, *BnaC01CRK5*, and *BnaA05CRK7*) and group C were preferentially expressed in sepals, filament, bud, and petals, whereas *BnaC03CRK1*, *BnaC04CRK7-1*, and *BnaA04CRK7* show higher expression in the stem, indicating the possible role of the group A and group C *BnaCRK* genes in flowering. In contrast, higher expression levels of the group B members, including *BnaA08CRK8*, *BnaC08CRK6*, *BnaA06CRK6*-*2*, *BnaC03CRK6*, *BnaC09CRK4*, and *BnaA09CRK4* were observed in siliqua. Interestingly, three members of the *BnaCRKs* (*BnaA06CRK6-1*, *BnaC04CRK7-2*, and *BnaC08CRK8*) with the highest expression in the root, showing relatively undetectable expression in other tissues, except *BnaC04CRK7-2*, whose mild expression was also observed in upper-stem-peel. To further verify the expression level, we employed qRT-PCR to determine the organ-specific expression of eight homologous *BnaCRKs* in different tissues including, leaf, stem, seed, flower, young flower, bud, young-leaf, and the root of the *B. napus* variant ZS11 (Figure 7). The transcriptional level of *BnaA03CRK1*, *BnaC04CRK3*, and *BnaA08CRK8* were upregulated in the flower and young flower. In contrast, the expression of *BnaA01CRK5*, *BnaA06CRK6-1*, and *BnaC04CRK7-2* showed the highest peak in the stem, root, and leaf, respectively (Figure 7). Additionally, the expression level of *BnaA05CRK2* has been observed at nearly the same level in all selected tissue (Figure 7; Appendix A). Together with microarray data, the transcriptional level of *BnaA03CRK1/BnaC04CRK3* and *BnaA06CRK6-1* were predicted in the flower and root, respectively, suggesting a positive correlation between qRT-PCR and the microarray data. Overall, the results from these datasets implied that the *BnaCRKs* expressed differently within the group and might regulate several growth and development processes of *B. napus*.

### 2.7. Prediction of Cis-Acting Elements in the Promoter Region of BnaCRKs

To further analyze the individual role of *BnaCRK* members and explore the regulatory mechanism governing *BnaCRK* in response to hormone signaling, development, and biotic and abiotic stresses, we isolated the 2000-bp promoter region of each *BnaCRK* gene to predict *cis*-acting elements. The outcomes disclosed that the 72 types of *cis*-elements were unevenly distributed in the promoter regions of the *BnaCRKs*. These elements were divided into four categories, light responsiveness, stress and defense-related, hormone responsiveness, and development-related (Figure 8), in which light responsiveness was the most significant number, represented by 21 types in the promoter region of the *BnaCRKs*, suggesting that BnaCRK transcriptional activity might be stimulated by the light condition (Appendix A). Overall, *BnaA06CRK6-2*, *BnaC04CRK7-1*, and *BnaC07CRK4* are increased in stress-defense-related, light responsiveness, and hormone-responsive *cis*-elements, respectively. In contrast, many *cis*-elements related to development responses are only detected in *BnaA08CRK8* (Figure 8). Nevertheless, a few *cis*-core elements were only seen in some *BnaCRK* family members. For instance, F-box (*cis*-element that mediates plant tolerance to abiotic and biotic stresses) was only found in *BnaA01CRK2*, *BnaC09CRK4,* and *BnaA09CRK4*. Furthermore, RY-element (*cis*-element involved in seed development), GCN4-motif (*cis*-element involved in regulating endosperm-specific regulation), Nodule site1 (*cis*-element that regulates the nodule-specific expression), and TGA-box (*cis*-element that plays a regulatory role during iron deficiency) were found in *BnaA03CRK1*, *BnaC08CRK6*, *BnaA09CRK4,* and *BnaC04CRK7-2*, respectively. AT1-motif (involved in light responsiveness), CAG-motif (cis-element involved in regulating salinity stress-related genes expression), Sp1 (cis-element showing a response to light), and ATC-motif (takes part in light response) were predicted only in *BnaC03CRK8*, *BnaC04CRK3*, *BnaA06CRK6-1,* and *BnaC07CRK4*. Similarly, ACTCATCCT sequence (*cis*-element for proline and hypoosmolarity-responsive expression), DRE-core (regulates cold and drought-responsive genes expression), MBS1 (MYB binding site involved in drought-inducibility), and GARE-motif (*cis*-element that takes part in gibberellin responses) were not found in all *BnaCRKs* except for *BnaA09CRK5*, *BnaC07CRK4*, *BnaC04CRK3*, and *BnaA06CRK6-2*, respectively. Additionally, defense and stress-responsive *cis*-elements, such as ARE (*cis*-acting element essential for anaerobic induction), ERE (ethylene-responsive *cis*-element), LTR (*cis*-element that shows regulatory response under low temperature), STRE (stress-response *cis*-element), W-Box (positive regulator of senescence-related genes), and WUN-motif, which is a wound responsive *cis*-element, were detected in all members of the *BnaCRK* gene family. Furthermore, hormone-responsive *cis*-elements, such as ABRE (*cis*-element involved in the abscisic acid responsiveness), TGA-element (*cis*-element involved in the auxin responsiveness), P-BOX (*cis*-element involved in the gibberellin responsiveness), TCA-element (*cis*-element that shows a response to salicylic acid) were perceived, in which the *cis*-elements affected by the auxin responses were found most common in all members of the *BnaCRK* gene family. Results from this analysis showed that the *BnaCRKs* contain different kinds of hormone-responsive, development-responsive, and stress-defense-related regulatory elements in their promoter regions, implying that *BnaCRKs* might regulate *B. napus* development in response to the different phytohormones treatments and stresses.

### 2.8. Expression Profile of the BnaCRKs under Abiotic Stresses and S. sclerotiorum Infection

Previous studies on *CRK* genes reported their fundamental role in the adaptation of plants to biotic and abiotic stresses by changing the cytosolic calcium concentration [19,58]. However, few studies report the response of *BnaCRKs* under biotic and abiotic stresses in *B. napus*. To explore the function of *BnaCRKs* under different environmental cues, including salinity, cold, freezing, flood, drought, heat, clubroot, and *S. sclerotiorum* stresses, we utilize the RNA-seq datasets (CRX052409, SRP190170, SRP231183, SRP277041, SRX9686328, and SRP311601) to investigate the expression pattern. As shown in Appendix A, we found that the majority of the *BnaCRKs* showed downregulation under cold, freezing, salinity, flood, drought, heat, clubroot, and *S. sclerotiorum* stresses, whereas multiple *BnaCRKs* were significantly induced by several stress treatments. For instance, *BnaC07CRK4*, *BnaA06CRK6-1*, and *BnaA06CRK6-2* were significantly induced by flood and salinity stress, while *BnaC04CRK3*, *BnaC04CRK7-2*, and *BnaA08CRK8* expression levels were increased in response to heat stress (Appendix A). Furthermore, we also found the expression level of *BnaC01CRK2, BnaC04CRK3, BnaC07CRK4, BnaA06CRK6-1, BnaC08CRK6*, and *BnaA08CRK8* were upregulated under *S. sclerotiorum* inoculation, while *BnaA04CRK3*, *BnaA09CRK4*, *BnaC01CRK5*, *BnaA03CRK8*, and *BnaC08CRK8* are predicted to show higher expression in response to clubroot disease.

To further understand their response in *B. napus* to different environmental cues, we analyzed the expression pattern of eight *BnaCRKs* in 18-day-old *B. napus* seedlings treated with drought (15% PEG 600), salt (200mM NaCl), freezing (−4 °C), and *S. sclerotiorum* by using qRT-PCR. We found that almost all the *BnaCRKs* expressions were significantly reduced in response to stresses except for *BnaA06CRK6-1* and *BnaA08CRK8,* whose expression was upregulated under drought and salt treatment (Figure 9). However, compared to freezing stress, drought and salt stress displayed a stronger response in the regulation of *BnaCRK* genes. Furthermore, the expression pattern of the candidate *BnaCRK* genes in response to host-pathogen *S. sclerotiorum* in *B. napus* was also investigated. At 24 h post-inoculation (24 hpi) of *S. sclerotiorum*, there was a significant increase in the expression levels of *BnaA03CRK1*, *BnaC04CRK3*, *BnaA06CRK6-1*, and *BnaA08CRK8*; however, at 48 hpi, the expression level was dramatically reduced (Figure 9; Appendix A). Interestingly, the *BnaA01CRK5* gene from group A showed a downregulation response at 12–48 h post-inoculation of *S. sclerotiorum*, which correlates with the RNA-seq data results (Appendix A; Appendix A). Overall, the expression levels of the *BnaCRKs* were significantly altered during 12–48 h of *S. sclerotiorum* infection. The results indicate that the members of the *BnaCRK* gene family display diverse expression patterns in response to abiotic and *S. sclerotiorum* treatment. However, *BnaA06CRK6-1* and *BnaA08CRK8* exhibit a stronger response, indicating that these two genes might direct plant resistance to a wide range of environmental stresses in *B. napus*.

### 2.9. Prediction of Potential MicroRNAs Targeting BnaCRKs Gene Family in B. napus

MicroRNAs (miRNAs) play a multifaced role in various aspects of plant development, including flowering time, hormone homeostasis, pattern formation, nutrient limitation, and the post-transcriptional regulation of targeted genes in response to stresses [59,60]. The members of the *CRK* gene family in maize and *Camellia siensis* L. are predicted to be post-transcriptionally mediated by the different miRNAs [61]. To better understand the transcriptionally regulatory role of *BnaCRKs* in response to stresses, we predicted the potential miRNA targets using a computational approach. In our analysis, we found that 19 *BnaCRKs* were putatively targeted by the 18 novel Bna-miRNAs, in which a single Bna-miRNA targeted 1 to 9 *BnaCRKs* (Figure 10; Appendix A). The six members from group A (*BnaC03CRK1, BnaA09CRK5, BnaC08CRK5, BnaC01CRK5*, *BnaC04CRK7-2,* and *BnaA05CRK7*) were predicted to be targeted by five novel Bna-miRNAs (Bna-miR156, Bna-miR9408, Bna-miR9557, Bna-miRN274, and Bna-miRN282), while only five members from group C (*BnaC05CRK2, BnaC08CRK6*, *BnaA06CRK6-1, BnaA06CRK6-2*, and *BnaC03CRK6*) were targeted by the three Bna-miRNAs (Bna-miRN277, Bna-miR408, and Bna-miR319). In addition, four known Bna-miRNAs (Bna-miR159, Bna-miR394, Bna-miRN284, and Bna-miRN285) were identified to target 15 members of the *BnaCRK* gene family (Figure 10). Moreover, the two members of group B (*BnaA04CRK3* and *BnaC04CRK3*) were targeted by Bna-miR2111, Bna-miR393, Bna-miRN273, Bna-miRN287, and Bna-miRN290, indicating that the different Bna-miRNAs might differently regulate the activity of the *BnaCRKs* transcripts. The binding energies between the Bna-miRNAs and their targets are between −7.9 and −26.34; the lower values indicate a higher degree of Bna-miRNA-target binding (Appendix A). However, except for Bna-miR394/*BnaC04CRK7-1*, *BnaA05CRK7*, *BnaC03CRK1*, Bna-miR9408/*BnaA05CRK7*, Bna-miR9557/*BnaC01CRK5*, and Bna-miRN285/*BnaA09CRK5*, the majority of the *BnaCRKs* transcripts are inhibited by the miRNAs cleavage in *B. napus*. In previous studies, Bna-miRNAs, such as (Bna-miR2111, Bna-miR408, Bna-miR9557, and Bna-miRN273) are reported to be involved in silique development [62], whereas Bna-miR156, Bna-miR159, Bna-miR395, Bna-miR319, Bna-miR394, and Bna-miR395 shows a response against *S. sclerotiorum* infection, salt, and drought stresses [62,63,64,65,66], in which the higher expression of Bna-miR395, Bna-miR319, Bna-miRN274, Bna-miR282, Bna-miRN284, Bna-miRN277, Bna-miR211, Bna-miRN290, and Bna-miRN287 were putatively observed under *S. sclerotiorum* infection in *B. napus* [67] (Figure 10), suggesting that the predicted Bna-miRNAs in this study might downregulate the respective targets of the *BnaCRKs* in response to multiple stresses.

### 2.10. SNP Polymorphism

To investigate the sequence polymorphism of the *BnaCRK* genes, we used the SNP data of 159 variants from the *B. napus* genome browser. Approximately 70% of high-quality SNPs were detected in the *BnaCRKs* (Appendix A). Nevertheless, the SNP density of the *BnaCRK*s was different within each subfamily. For instance, 71% of SNPs were identified in Group A and B, while Group C holds an average of 50% of SNPs. Moreover, it has also been observed that the number of SNPs was relatively higher in the *B. napus* CC genome rather than in the AA genome (Appendix A). Furthermore, we have also predicted the distribution of SNPs on each member of the *BnaCRK* gene family. As detailed in Appendix A, we found that the SNP distribution on exons ranged from 22 to 36%, in which the *BnaA08CRK8* exon-region contains the highest number of SNPs (81). In contrast, the SNP distribution in the intron region varied from 28 to 35%, whereas only one SNP was found in the intron of *BnaC09CRK4*, suggesting that the sequence variation in the *BnaCRKs* might contribute to their different expression pattern in response to biotic and abiotic stresses (Appendix A).

## 3. Discussion

Rapeseed (*Brassica napus*) is an essential oil crop throughout the world. Nevertheless, several environmental factors, including drought, higher salinity, flood, and pathogen infection, cause a significant annual economic loss. Improving genetic tolerance to such harsh conditions is the preferred approach in many crops [68,69,70,71,72]. Several studies have been done to characterize and determine the important gene families in response to several environmental stresses in *B. napus* [73,74,75,76]. Despite this, the *CRK* gene family, which has an indispensable role in plant tolerance to environmental cues, has not been identified in *B. napus*. *CRKs* comprise a single-transmembrane domain, an extracellular domain, and a Ser/Thr protein kinase domain [13]. It has been reported that a group of the cystine-rich-receptor-like kinase domains of the *CRK* encoding the C-8X-C-2X-C motif was induced by pathogen infection, oxidative stress, and abiotic stresses [14,15,16]. The *CRK* gene family has been identified and characterized in many plants. For instance, *O. sativa* contains (36), *Gossypium barbadense* (30), *S. lycopersicum* (6), *Cucumis melo* L. (7), and *Glycine max* contains (91) members of the *CRK* gene family (Table 1). Despite their large number, only a few have been functionally described. For instance, previous studies on *Arabidopsis* have reported the overlapping function of *AtCRK6* and *AtCRK7* in mediating the response to extracellular reactive oxygen species (ROS) [33], while the overexpression of *CRK5* significantly influences the expression of isochorismate synthase 1 (*ICS1*), and pathogenesis-related protein (*PR*) to increase plant tolerance to *Pseudomonas syringae* infection, and also shows a positive response against salt tolerance [77,78]. Furthermore, a recent study has reported the direct interaction of *RLCK* with *AtCRKs*, which eventually induces plant immunity against pathogen attacks [79]. Additionally, the *TaCRK1* gene isolated from stems of wheat was reported to show enhanced expression in response to *Rhizoctonia cerealis* [35]. Moreover, a semi-dominant mutation of the *CRK* gene (*als1*) in rice resulted in an improved response to rice leaf blast disease [80]. However, the number of *CRKs* in *B. napus* remains ambiguous. Our study systematically characterized the *BnaCRK* gene family based on their structure, phylogenetic relationship, conserved domain, protein-protein interaction, and expression pattern in different tissues and in terms of their response to freezing, drought, salinity, and *S. sclerotiorum* treatment. Insights from these results provide a better understanding of the *CRK* gene family in *B. napus* and provide a genetic basis for advancing *B. napus* breeding.

We have found that *B. napus* contain 27 putative members of the *CRK* gene family, divided into three groups based on the phylogenetic relationship among *B. rapa, A. thaliana*, *B. nigra, B. juncea*, and *B. oleracea* (Figure 1). Group A and B contain a more diverse number of *CRK* genes than group C, supporting the notion that the *CRK* genes in the denoted plant species are evolved mainly by the members in group A and C. Compared to the *CRK* genes in *B. rapa* (15), *B. oleracea* (11), *B. nigra* (14), and *B. juncea* (24), *B. napus* contain a higher number of *CRK* genes (Figure 1), which might be due to the larger genome of the *B. napus* and multiple incidences of gene duplication. All 27 members of the *BnaCRK* gene family hold the conserved N-terminal protein between each other (Appendix A; Figure 2). For instance, members of *BnaCRK6* from group B contain only one exon without any intron, while the rest contain 2 to 11 exons with 10 introns (Appendix A). In addition, motif 16 was only predicted in group C and *BnaCRK6*, suggesting the diversity of *BnaCRK* functions within the same subfamily (Figure 2). It has been reported that the diversity of gene functions within a subfamily is due to a mutation in the amino acid site [81,82]. Type I and Type II divergence analysis was performed to predict whether mutations in the amino acid sites were the cause of the functional divergences between the members of the *BnaCRK* gene family. As detailed in Table 2, we observed the different theta *ϴ* values between each group, indicating the significant divergence between *BnaCRKs*. Moreover, 20 critical amino acid sites were predicted within group A/B, group A/C and group C/B, in which 5 amino acid sites (3I, 100K, 104P, 113R, and 114A) were relatively conserved in both Type I and Type II divergence, suggesting that the different evolutionary rate at these sites might evolve *BnaCRK* genes to novel functions after divergence (Table 2). Additionally, these amino acid sites were mainly allocated on the protein kinase domain (Figure 3), indicating the vulnerability of the N-terminus PKINASE domain to positive selection during the evolution of the members of the *BnaCRK* gene family.

*B. napus* is an angiosperm crop in which genome duplication leads to an increase in whole genome genes [42]. Our synteny analysis predicted that segmental duplication plays a vital part in the expansion of *CRK* genes across all *B. napus* chromosomes, which might explain the higher number of *CRK* genes in *B. napus* (Figure 4A). Among 42 segmental duplication pairs (Figure 4B), *BnaCRK1/BnaCRK7* and *BnaCRK2/BnaCRK8* are considered putatively sufficient for mediating calcium signals in response to different environmental cues. Additionally, 40, 68, 70, and 71 orthologous gene pairs were predicted among *A. thaliana/B. napus, B. oleracea/B. napus, B. rapa*/*B. napus*, and *B. n*igra/*B. napus*, respectively (Figure 5; Appendix A). The formation of these orthologous pairs among *B. napus* and denoted *Brassicaceae* species might be related to the functional diversification and evolutionary mechanism of the *BnaCRKs*.

Tissue-specific expression analysis provides essential insights to predict gene-specific function. In this study, we found that the *BnaCRKs* expressed ubiquitously in the root, stem, flower, young flower, bud, and leaf (Figure 7; Appendix A). However, some of the *BnaCRKs* are expressed highly in dissimilar tissue, indicating the tissue-specific function of some *BnaCRKs*. For instance, *BnaC03CRK1* and *BnaC04CRK3* were expressed exclusively in flowers, and their homologous pairs in *A. thalian* were exhibited to participate in pollen tube formation [83]. Consistent with this, it was reported that the *CRK* genes were involved in several regulatory processes during the different stages of plant development [20,37,57,84] and in regulating different phytohormones [85]. For instance, the tomato *CRK* gene (*LeCRK1*) was determined to take part in fruit ripening [17], while an *A*. *thaliana CRK* gene (*AtCRK5*) was noted to participate in the root gravitropic response by regulating auxin transport [86]. In *A. thaliana,* a *crk* T-DNA insertion mutant displays an abnormal root phenotype [32]. Intriguingly, our study found that the *BnaA06CRK6-1* was expressed highly in the root, indicating that the different *BnaCRK*s might mediate different signaling pathways of plant development (Figure 7).

In our protein-protein interaction analysis, we have perceived that all *BnaCRKs* showed strong interaction with stress-induced and development-related transcriptional factors (Figure 6A). For instance, the members of BnaCRK1 and BnaCRK5 from group A interacted with HSF1, ARCK1, PP2C, and RLKs (Appendix A), which positively regulate the plant development in response to various environmental stresses [27,28,55,56,86] (Figure 6B). Several studies have reported the function of *CRKs* in plant defense under different stresses [87,88,89,90]. In these studies, several members of the *CRK* gene family were transcriptionally modified in response to abiotic stresses. In our research, most of the *BnaCRKs* showed a downregulation response to freezing, salt, and drought, except for *BnaA06CRK6-1* and *BnaA08CRK8*, whose expression was upregulated to drought, salt, and *S. sclerotiorum* stress (Figure 9). Furthermore, a higher number of stress-defense-related, hormone-responsive, and development-related *cis*-elements were predicted in the promoter region of *BnaA06CRK6-1* and *BnaA08CRK8* (Figure 8; Appendix A). Additionally, to gain further insights into the post-transcriptional modification of the *BnaCRKs* under different stresses, we have predicted that the 19 *BnaCRKs* were putatively targeted by 18 novel Bna-miRNAs (Figure 10), in which *BnaA06CRK6-1*, whose expression was significantly stimulated by salt, drought and *S. sclerotiorum* stress, was putatively targeted by Bna-miRN277, Bna-miR408, and Bna-miR319, which are found to participate in gene post-transcriptional regulation under biotic and abiotic stresses [62,63,64,65,66], indicating the potential function of the *BnaCRKs* in response to multiple stresses in *B. napus.* Our findings give an insight into the functional divergence of the *CRK* genes in *B. napus*, and pave the way to an exploration of the biological role of these genes in *B. napus* by further experiments.

## 4. Materials and Methods

### 4.1. Identification, Phylogenetic Analysis, and Structural Characterization of B. napus CRK Family

To identify the *CRK* gene family in *B. napus*, we first obtained the eight *AtCRKs* (*AtCRK1, AtCRK2, AtCRK3, AtCRK4, AtCRK5, AtCRK6*, *AtCRK7*, and *AtCRK8*) protein sequences from the *A. thaliana* genome database (TAIR: http://www.arabidopsis.org; accessed on 15 March 2021) [91] and utilized the BLASTp function in the *B. napus* genome browser (BnPIR: http://cbi.hzau.edu.cn/bnapus; accessed on 15 July 2021) [43,44] and (GENOSCOPE: https://www.genoscope.cns.fr/brassicanapus; accessed on 20 July 2021) [42] to obtained the BnaCRK proteins sequences. Additionally, *CRK* protein sequences from *B. nigra*, *B. juncea*, *B. rapa,* and *B. oleracea* were also gained using the *AtCRKs* as a reference in the *Brassica* database (BRAD: http://brassicadb.cn accessed on 25 April 2022) [92], and Phytozome: https://phytozome-next.jgi.doe.gov/; accessed on 20 April 2022 [93]. All the retrieved *CRK* sequences with 80% similarity were subjected to domain prediction by employing the Pfam: http://pfam-legacy.xfam.org/search/null; accessed on 20 May 2022 [94], CDD: https://www.ncbi.nlm.nih.gov/Structure/bwrpsb/bwrpsb.cgi; accessed on 20 June 2022 [95], and ScanProsite: https://prosite.expasy.org/scanprosite/; accessed on 20 June 2022 [96] databases. Sequences with shorter amino acid (<90) and lacking matching domains were removed, and the remaining 99 CRK protein sequences from *B. napus*, *A. thaliana, B. nigra*, *B. juncea*, *B. rapa*, and *B. oleracea* were aligned to verify the conserved domain. The phylogenetic tree was constructed using the NJ (Neighbor-joining) function with the bootstrap test for 1000 replicates in MEGA 11 [97] and visualized using iTOL [98] software. To explore the CRK gene structures, we manually aligned each gene genomic and coding sequence and used the FGENESH and GSDS [99] programs for visualization. Furthermore, the N-terminal myristoylation sites, cellular location, and physiochemical properties of the CRK proteins were identified using the Myristoylator [100], Plant-mPLoc [101], and ProtParm [102] tools, respectively.

### 4.2. Motif Composition, Genomic Distribution, Site-Specific Assessment, and Synteny Analysis of CRK Gene Family in B. napus

Conserved motifs were identified by uploading the protein sequences of BnaCRKs and AtCRKs into the MEME: http://meme-suite.org/tools/meme; accessed on 20 May 2022 [103] server, with the maximum number of motifs adjusted to 20. To locate the *BnaCRKs* on the genome of *B. napus*, we acquired the genome information from the *B. napus* database BnPIR and visualized it using the TBtools [104] software. Furthermore, the MCScanX toolkit [105] was employed to classify segmental and tandem duplication events in the *BnaCRKs*. Additionally, the Dual Synteny plotter function in TBtools was used to construct the relationship between the *BnaCRKs* and *CRKs* from the denoted genomes. The selective pressure of the *BnaCRKs* was predicted utilizing the EasyCodeML [54] script and Selecton Server [106].

### 4.3. Functional Divergence and Three-Dimensional Architecture of CRK in B. napus

We utilized the DIVERGE 3.0 [48] program to identify the type I and type II functional divergence between *BnaCRK* groups, with the Bayesian posterior probability (Qk) score set to 0.9. Furthermore, to present the three-deletional architecture of the BnaCRKs, a candidate BnaA03CRK1 protein sequence was uploaded to the I-TASSER [107] database and annotated by the PyMOL program.

### 4.4. BnaCRKs-Target Interaction, Cis-Acting Elements, and Micro-RNA Target Site Prediction

BnaCRKs target interactions were investigated by submitting the AtCRKs orthologs protein sequences to the STRING [108] database, and the chart was generated by employing the Cytoscape tool. Additionally, the KEGG pathway was drawn utilizing the KEGG enrichment function in TBtools. To identify the Bna-miRNA interaction with *BnaCRKs*, we obtained the novel Bna-miRNAs from PNRD [109] and miRbase [110] database and aligned them with the 27 members of the BnaCRK gene family using the psRNAtarget [111] server. Furthermore, the expression profiles of the predicted Bna-miRNAs under *S. sclerotiorum* infection were acquired from the RNA-Seq dataset (SRP075341), and the network was drawn using Cytoscape. To reveal the *cis*-acting elements, a 2.0 kb upstream promoter sequence from each *BnaCRK* was collected and analyzed by employing the PlantCARE [112] server.

### 4.5. B. napus Growth Condition and Expression Analysis of Selected BnaCRKs under Different Stresses

*B. napus* cultivar Zhongshuang 11 (ZS11) was germinated in the growth room at 20 ± 5 °C, 16 h light/8 h dark at 50 µmol/m2/s light intensity with 70% relative humidity. Tissues from the plant, including flower, root, seed, stem, young flower, young leaf, bud, and leaf, were collected and kept at −80 °C for tissue-specific gene expression analysis. For stress treatment analysis, 18-day-old *B. napus* seedlings were exposed to freezing (−4 °C), drought (15% PEG 600), and salinity (200mM NaCl) treatments for 24 h. Three biological replicates were collected and instantly stored at –80 °C. For fungi inoculation, the mycelia of *S. sclerotiorum* was inoculated on a potato dextrose agar (PDA) at 25 °C for two days. PDA plugs of actively growing *S. sclerotiorum* (8 mm in diameter) were placed on the surface of fully-grown four-week-old *B. napus* leaves. The three biological replicates were collected at three different time points (0, 12, and 24 h) after inoculation and instantly frozen in liquid nitrogen for RNA isolation. Total RNA was extracted, and cDNA was synthesized according to our previous report [113]. For qRT-PCR analysis of the *BnaCRK* expressions, we used the method mentioned in our previous study [74]. The primers utilized in the qRT-PCR analysis are detailed in Appendix A.

### 4.6. RNA-Seq and SNP Distribution Analysis of BnaCRKs

The transcriptomic data subjected to different stress treatments, including heat, drought, flood, salinity, cold, freezing, clubroot, and *S. sclerotiorum,* were retrieved from the RNA-seq data sets (SRP277041, SRP231183 [114], CRX052409, SRP190170 [115], SRP297988, and SRP311601), respectively to explore the expression patterns of the *BnaCRKs*. The differential expression was measured by the DSEeq2 package in R-studio, and then the values were calculated by the log2 fold change (log_2_FC) method. To analyze the natural variation of *B. napus CRK* genes genomic sequences, the SNPs were determined in the 159 cultivars of the *B. napus* extracted from the BnPIR genome browser. The SNPs with a missing rate below <50% were used for SNP distribution onto the genomic structure of *BnaCRKs*.

### 4.7. Statistical Analysis

Data reported in this study were the average of three replicated treatments and each treatment consisted of six-to-eight seedlings. The transcriptional level assayed by qRT-PCR was observed using the 2^−∆∆Ct^ method as described in our previous study [116], and the subsequent statistical analysis was performed by standard deviation and one-way analysis of variance (ANOVA) followed by the Dunnett’s *t*-test to find the significant difference of all stress treatments with the controls if the ANOVA is significant at *p* < 0.05. The statistical analyses were evaluated using the GraphPad Prism 8.0 program. The Pearson correlation between qRT-PCR and RNA-seq datasets was performed using the Pearson correlation coefficient test in R, and the correlation was considered significant at *p* < 0.001. 

## 5. Conclusions

In this report, we identified the 27 members of the *CRK* gene family in the *B. napus* genome that were unevenly distributed between the 14 chromosomes. The gene structure, motif distribution, functional divergence, gene duplication, *cis*-elements, miRNA target prediction, protein-protein interaction, and expression response to heat, drought, flood, cold, freezing, clubroot, and *S. sclerotiorum* were thoroughly investigated in *B. napus*. We observed that all the *BnaCRKs* expressed ubiquitously in several tissues and showed increased expression response to *S. sclerotiorum* infection, suggesting the positive role of *BnaCRKs* in the stress resistance of *B. napus*. Our work provides valuable insights into the evolution and functional divergence of the *BnaCRKs* in *B. napus* and displays the possibility of *BnaCRKs* for genetic improvement of the *B. napus* breeding strategy.

## Figures and Tables

**Figure 1 ijms-24-00511-f001:**
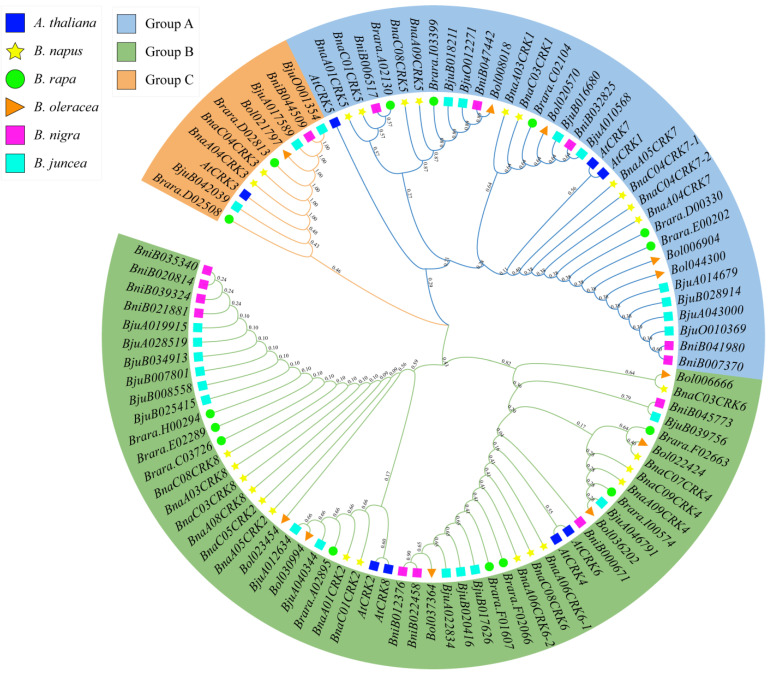
Phylogenetic tree analysis of *CRK* genes from *A. thaliana* (*At*), *B. napus* (*Bna*), *B. oleracea* (*Bol*), *B. rapa* (*Bra*), *B. juncea* (*Bju*), and *B. nigra* (*Bni*). The *CRK* genes are categorized into three groups (A, B, and C) and indicated as different colors.

**Figure 2 ijms-24-00511-f002:**
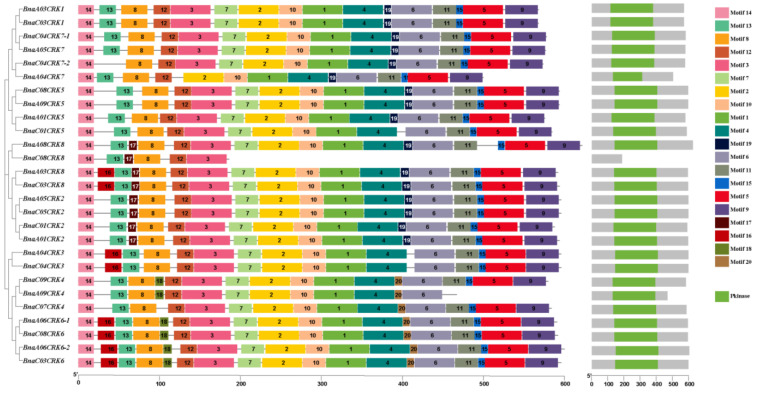
Conserved motif structure of the *BnaCRK* gene family. Different colors and numbers represented the 20 conserved motifs of the *BnaCRKs*. The logo of each motif was plotted in the supplementary Appendix A. The N-terminal PKINASE domain is indicated as a green-colored block.

**Figure 3 ijms-24-00511-f003:**
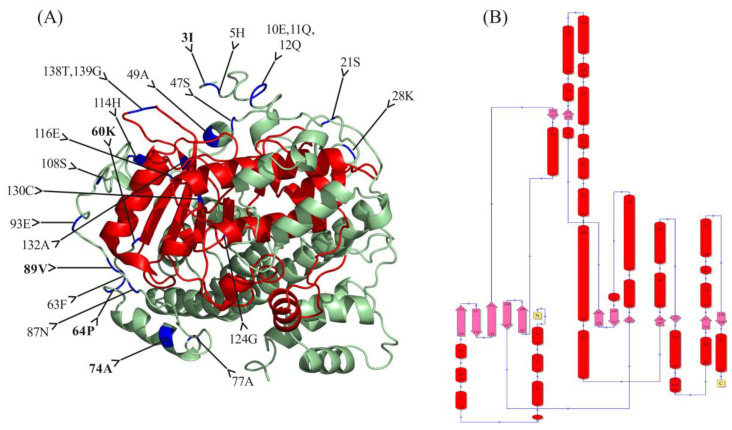
Three-dimensional putative architecture of BnaC01CRK1. (**A**) The N-terminal PKINASE domain is highlighted in red and blue shows the specific amino acid sites that are perceived to be involved in the functional divergence of the *BnaCRKs*. Amino acid sites that are predicted in both type I and type II functional divergence are indicated in bold letters. (**B**) The putative 2D structure of BnaC01CRK1. The pink arrows represent the strands, and the round cylinders indicate the helix.

**Figure 4 ijms-24-00511-f004:**
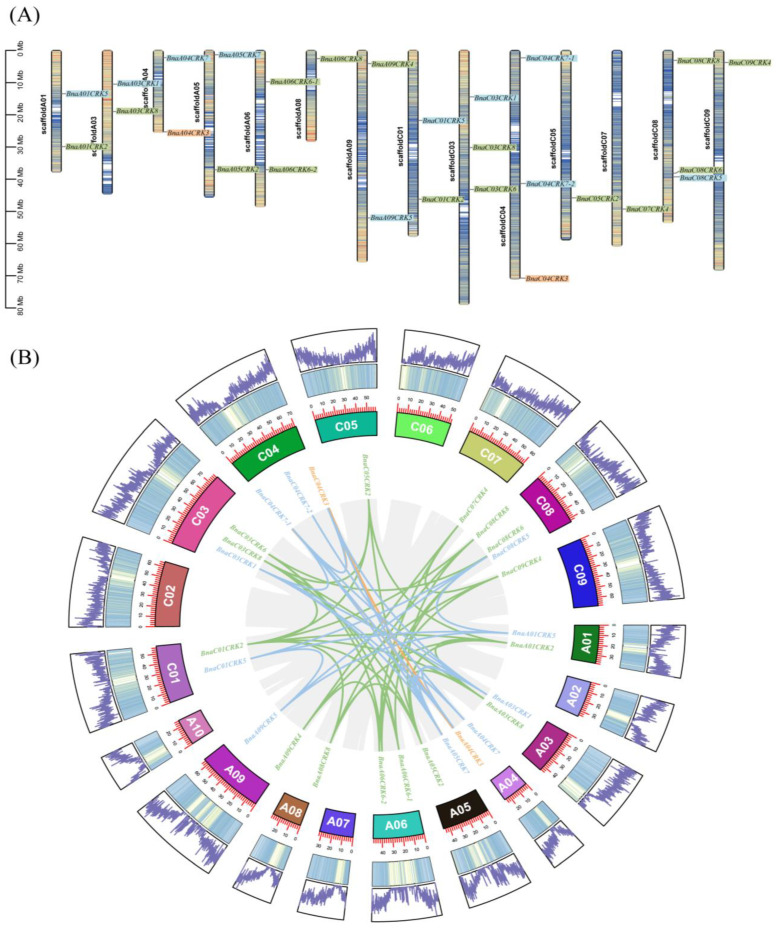
Chromosomal position and intrachromosomal relationship of the *BnaCRKs*. (**A**) Position of 27 putative *BnaCRKs* on the *B. napus* chromosomes. The same highlighted color indicates the genes from the same group. The scale bar shows an 80 Mb chromosomal distance. (**B**) Duplication of the *BnaCRKs* in *B. napus*. The lines with the same color are the members of the same *BnaCRK* group and indicate a segmental duplication pair. Synteny blocks of *the B. napus* genome are represented by gray lines in the background. The *B. napus* chromosomal gene density profile is shown by the outside circle, and the chromosome number is shown as a separate color box with a scale size in Kb.

**Figure 5 ijms-24-00511-f005:**
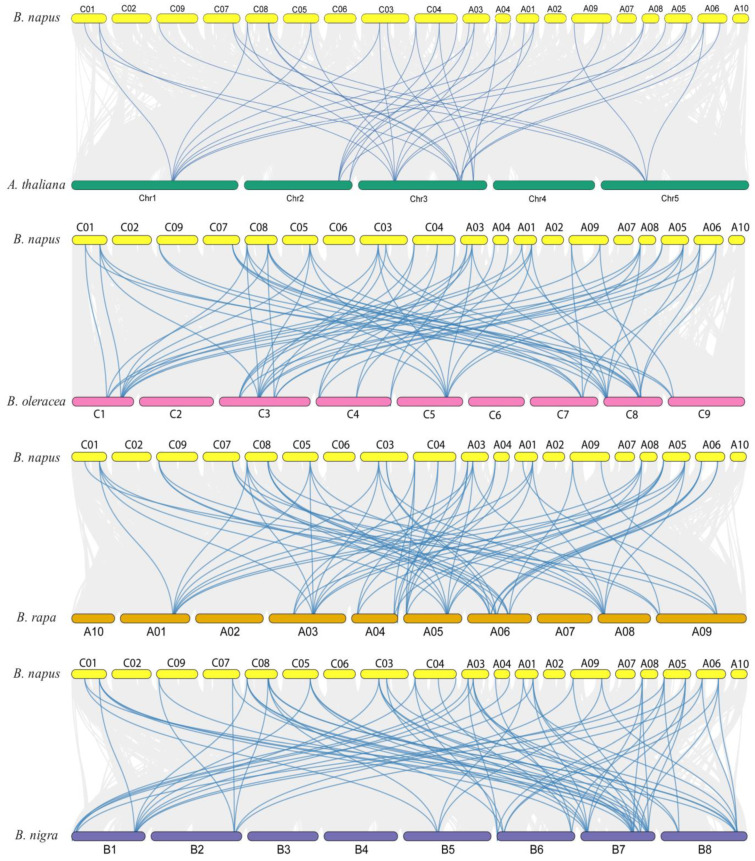
Collinear pair prediction of *BnaCRKs* among *B. napus* and four representative plant species. Gray lines in the background represent the collinear blocks within *B. napus*, *A. thaliana*, *B. oleracea*, *B.rapa*, and *B. nigra*, whereas blue lines indicate collinear pairs of *BnaCRK*s.

**Figure 6 ijms-24-00511-f006:**
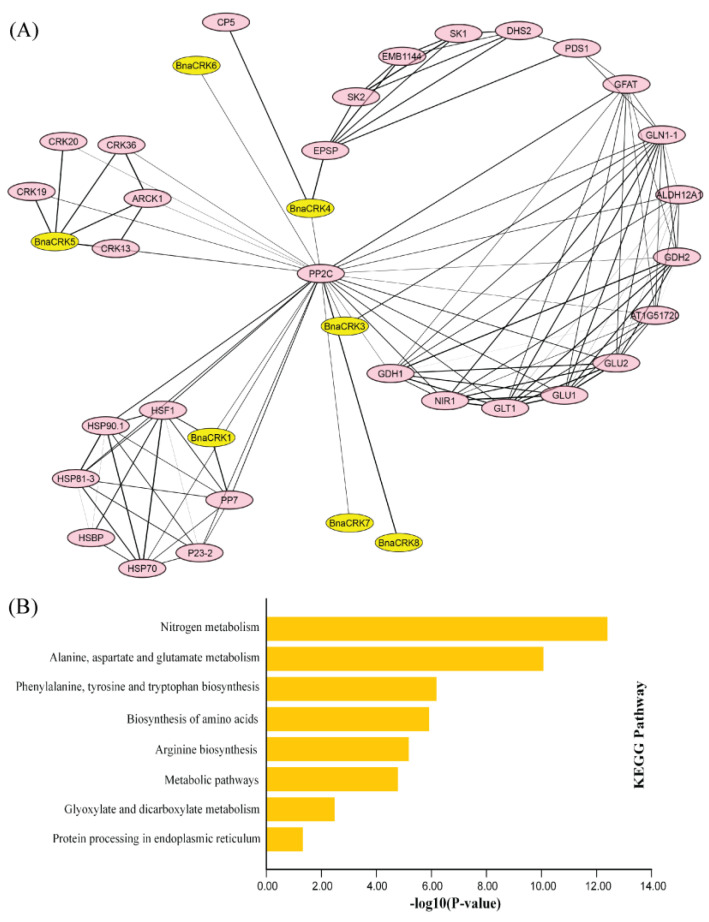
Protein interaction network of the BnaCRKs based on the *A. thaliana* orthologs. (**A**) Stronger interactions are shown by thicker lines. (**B**) KEGG pathway analysis of the BnaCRKs interacting targets.

**Figure 7 ijms-24-00511-f007:**
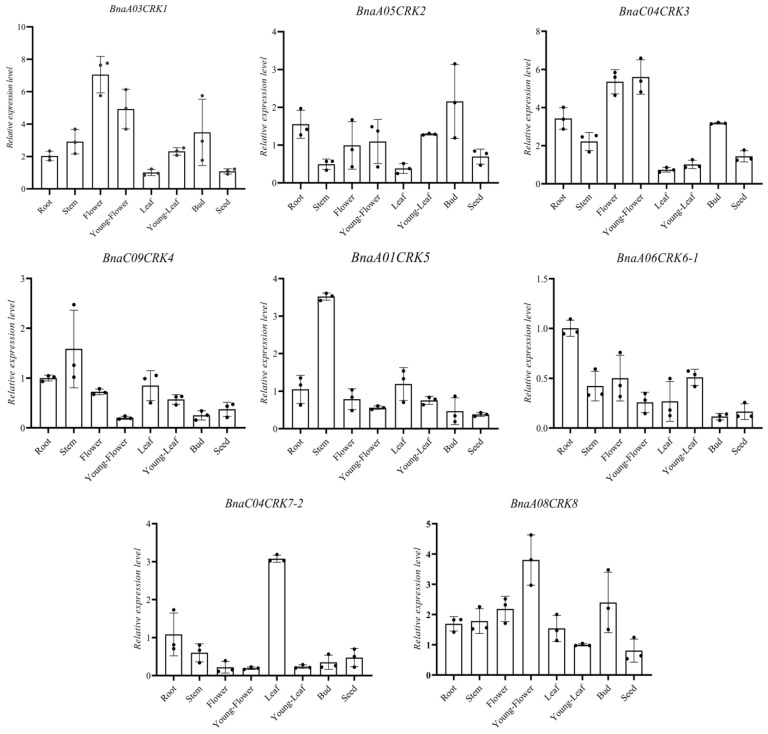
Tissue expression profile of the candidate *BnaCRKs*. The *B. napus Actin* (gene ID: XM013858992) was used to adjust the expression level of the selected *BnaCRKs*. The x-axis shows the name of the tissues. The error bars on the y-axis represent the data from qRT-PCR, which is the mean of three biological and technical replicates (Appendix A).

**Figure 8 ijms-24-00511-f008:**
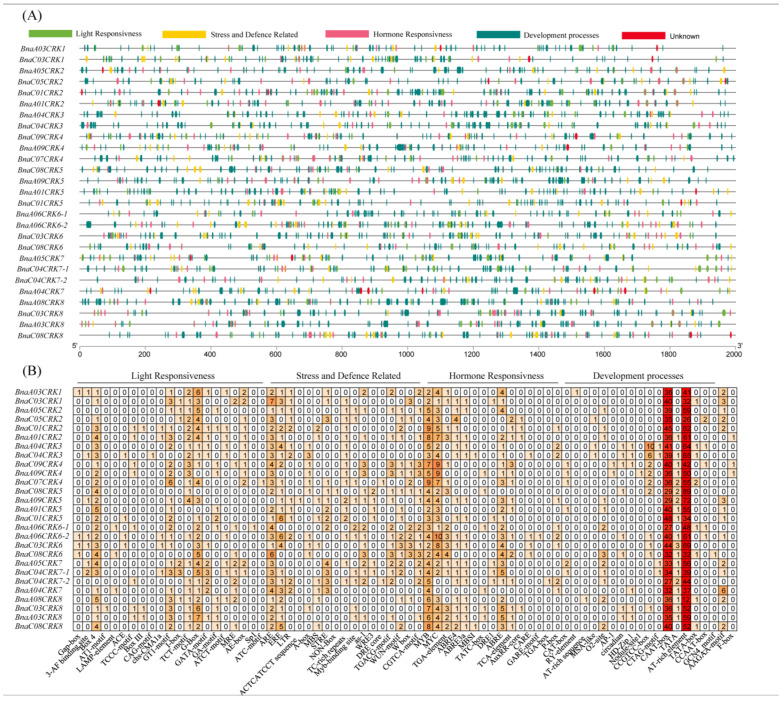
*cis-*acting regulatory elements analysis of the BnaCRKs. (**A**) Distribution of light-responsive, stress and defense-related, hormone-responsive, and development-related cis-core elements in the promoter region of each member of the *BnaCRK* gene family. (**B**) Total number of each *cis*-acting element in the promoter region of the *BnaCRKs*.

**Figure 9 ijms-24-00511-f009:**
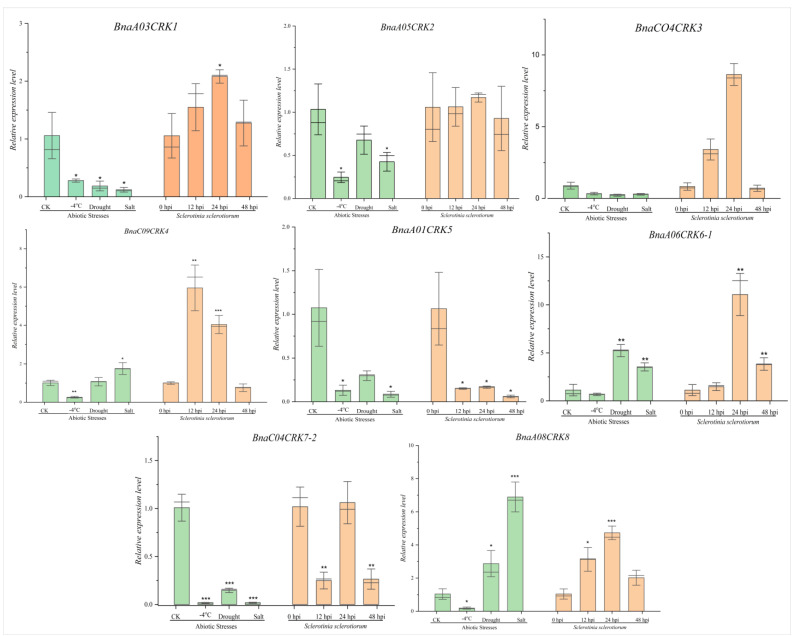
Relative expression pattern analysis of *BnaCRKs* in response to multiple stresses. CK represents the control group. The *B. napus Actin* (gene ID: XM013858992) was used to adjust the expression level of the selected *BnaCRKs*. Expression data from three independent biological replicates with standard error ± (SE) were displayed on the *y*-axis, and the *x*-axis represents the different treatments (mentioned in Appendix A). Significant differences are denoted by asterisks on the vertical bar at * *p* < 0.05, ** *p* < 0.01, *** *p* < 0.001.

**Figure 10 ijms-24-00511-f010:**
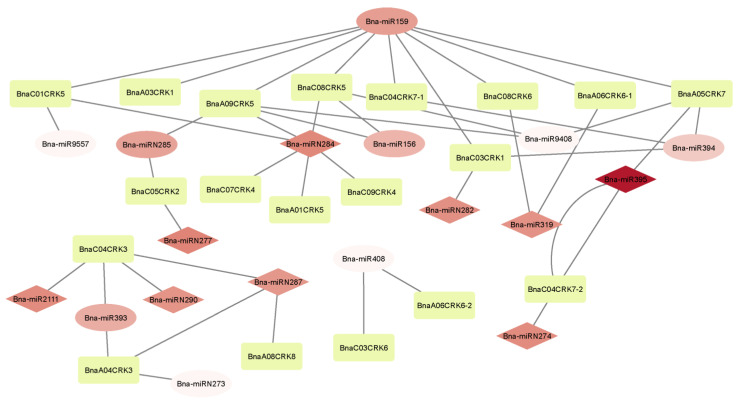
Interaction network of Bna-miRNAs with its targets. Bna-miRNA, whose expression was higher in response to *S. sclerotiorum*, is displayed in a diamond shape, while the yellow nodes represent the *BnaCRKs*.

**Table 1 ijms-24-00511-t001:** Identification of the *CRK* gene family in other plant species.

Plant Name	Common Name	Number of *CRKs*	Chromosome Number	Genome Size in Mb	References
*Gossypium barbadense*	Cotton	30	26	2500	[19]
*Arabidopsis thaliana*	Thale cress	8	5	135	[36]
*Solanum lycopersicum* L.	Tomato	6	12	950	[20]
*Malus domestica*	Apple	36	17	750	[22]
*Oryza sativa*	Rice	36	12	430	[21]
*Hevea brasiliensis*	Rubber plant	9	36	991	[37]
*Cucumis melo* L.	Melon	7	24	375	[38]
*Glycine max*	Soybean	91	10	1150	[39]
*Medicago truncatula*	Legume	6	16	430	[40]
*Sorghum bicolor*	Great millet	38	10	730	[21]
*Capsicum annuum* L.	Bell pepper	22	12	3058	[41]

**Table 2 ijms-24-00511-t002:** Functional divergence prediction between groups of the *BnaCRK* gene family.

Cluster I	Cluster II	*ϴ*I (Type I)					*ϴ*II (Type II)			
		Coefficient *ϴI*	S.E.*ϴ*I	LRT	Qk > 0.7	Critical Amino Acids	Coefficient *ϴII*	S.E.*ϴ*II	Qk > 0.7	Critical Amino Acids
**Group A**	Group B	0.126034	0.106769	0.917987	4	**60K**, **64P, 74A**, 115Y	0.077776	0.119825	29	**3I,** 5H, 10E, 11Q, 12Q, 15Q, 16S, 19V, 21S, 23Q, 28K, 40L, 46P, 47S, 49A, 58I, **60K**, 62P**, 64P, 74A,** 77A, 108S, 114H, 116E, 124G, 130C, 132A, 138T, 139G
**Group A**	Group C	−0.228125	0.027807	nan	2	**89V**, 117E	−0.147697	0.14611	21	5H, 36A, 37K, 38S, 41F, 44Y, 47S, 49A, 63F, 68S, 69S, 74A, 77A, 87N, **89V,** 93E, 108S, 132A, 138T, 139L
**Group B**	Group C	0.207912	0.141844	1.274511	5	**3I**,20S,39S,**89V**,117E,	0.082203	0.129724	42	**3I,** 5H, 9I, 10E, 11Q, 12Q, 21S, 22Q, 24S, 26V, 28K, 29D, 36A, 37K, 38S, 41F, 44Y, 47S, 63F, **64P**, 65A, 67A, 68S, 69S, 72L, **74A,** 76K, 77A, 78P, 80P, 87N, **89V**, 90A, 93E, 108S, 114H, 116E, 124G, 130C, 132A, 138T, 139L

## Data Availability

Not applicable.

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
