# Peer review of "Functional Characterization of the Cystine-Rich-Receptor-like Kinases (CRKs) and Their Expression Response to Sclerotinia sclerotiorum and Abiotic Stresses in Brassica napus"

_ijms, 2022, doi:10.3390/ijms24010511_

Round 1
Reviewer 1 Report
The presented manuscript is extensive and detailed with a lot of valuable information about CRKs identified in Brassica napus, their phylogenetic relationship, and their involvement in the stress signaling pathway.
However, I would suggest the authors consider changing its title. In the Results section, there are multiple stressors (drought, cold, salt) applied in order to determine CRKs expression patterns during a stress response, so it is not clear why the authors highlighted Sclerotinia sclerotium treatment in the manuscript title.
Also, verification of papers cited in this manuscript is necessary.
Other suggestions and comments are given in the pdf file of the revised Manuscript.

Author Response
Response to the Reviewer
Thank you for giving us the opportunity to submit a revised draft of our manuscript. We appreciate the time and effort that you dedicated to providing feedback on our manuscript and are grateful for the insightful comments and valuable improvements to our paper. Please see below, in blue, for a point-by-point response. We have also highlighted the changes within the manuscript.
Comment 1; The presented manuscript is extensive and detailed with a lot of valuable information about CRKs identified in Brassica napus, their phylogenetic relationship, and their involvement in the stress signaling pathway. However, I would suggest the authors consider changing its title. In the Results section, there are multiple stressors (drought, cold, salt) applied in order to determine CRKs expression patterns during a stress response, so it is not clear why the authors highlighted Sclerotinia sclerotium treatment in the manuscript title.
Response; Thank you for pointing this out, We authors chose to submit our work in the special issue "The Gene, Genomics, and Molecular Breeding in Cruciferae Plants" of the International Journal of Molecular Sciences. This special issue aims to improve the quality of oil crops and disease resistance including Sclerotinia sclerotiorum. Thus, to attract readers to the special issue, we decided to highlight the Sclerotinia sclerotiorum as the main treatment. However, as per the reviewer suggestion, we decided to change the title to “Functional characterization of the cystine-rich-receptor-like kinases (CRKs) and their expression response to Sclerotinia sclerotiorum and abiotic stresses in Brassica napus” please check in the revised manuscript.
Comment 2; Also, verification of papers cited in this manuscript is necessary. Other suggestions and comments are given in the pdf file of the revised Manuscript.
Response; Thank you for your suggestion, we have carefully revised the manuscript and tried our best to verify each reference.
Comment 3; Please verify this reference. Nothing about Alternaria brassicae in cited manuscript.
Response; Thank you for pointing this out, we have added the reference related to Alternaria brassicae. Please check lines 34-36 and reference 3 in the revised manuscript.
Comment 4; Nothing about 1000 copies of RLKs in the Oryza sativa genome. Please correct.
Response; Thank you for pointing this out, we have added the reference related to Oryza sativa RLKs. Please check line 43 and reference 9 and 10 in the revised manuscript.
Comment 5; Please check if 21 is the correct reference for this sentence.
Response; Thank you for pointing this out, we think that the work and core idea of reference 21 [1] and 22 [2] are the same, and in both studies, the authors have indicated the possible role of CRK5 and CRK4 in leaf growth and plant defense responses. Therefore, we believe that it is correct. To make it clearer, we decide to rephrase the sentence. Please check lines 63-65 and reference 25 and 26 in the revised manuscript.
Comment 6; Please check this. Is CRK3 calmodulin-binding protein kinase 3?
Response; Thank you for pointing this out, according to the Arabidopsis thaliana genome database (TAIR: http://www.arabidopsis.org) the other name of calmodulin-binding protein kinase 3 (AtCBK3) is AtCRK1, which we have corrected it, please check line 66-67 in the revised manuscript.
Comment 7; BnaA09CRK4 is missing its yellow star.
Response; Thank you for pointing this out, we have corrected it.
Comment 8; Please check reference 29
Response; Thank you for pointing this out, we have removed the unrelated reference from the sentence. Please check lines 314-315 in the revised manuscript.
Comment 9; Please specify what the CK value on the graphs stands for.
Response; Thank you for pointing this out, CK represents the control group, we have specified it in the figure 9 legend please check.
Comment 10; please check reference 68
Response; Thank you for pointing this out, we have corrected it please check the new reference 40 and lines 447-448 in the revised manuscript.
Comment 11; ...tissue-specific gene expression analysis.
Response; Thank you for pointing this out, we have corrected it. Please check lines 549 in the revised manuscript.
References
- Chen, K.; Du, L.; Chen, Z., Sensitization of defense responses and activation of programmed cell death by a pathogen-induced receptor-like protein kinase in Arabidopsis. Plant molecular biology 2003, 53, (1), 61-74,
- Chen, K.; Fan, B.; Du, L.; Chen, Z., Activation of hypersensitive cell death by pathogen-induced receptor-like protein kinases from Arabidopsis. Plant molecular biology 2004, 56, (2), 271-283,

Reviewer 2 Report
The manuscript “Functional characterization of the cystine-rich-receptor-like ki- 2 nases (CRKs) and their expression response to Sclerotinia sclerotiorum in Brassica napus” is well structured and the aim of the research work is relevant. I have the following concerns regarding the current manuscript that authors should consider while revising their manuscript.
1. The authors should abbreviate the genus name after the first use in the text. Like Sclerotinia sclerotiorum, Solanum lycopersicum, and Oryza sativa. Please check throughout the manuscript.
2. Statistical analysis should be included in "Materials & Methods". In what software the statistical analysis was carried out? How Statistical Significance was determined? What statistical test was used?
3. In Figure 9 graphs (x-axis), Sclerotinia sclerotiorum should be italic.
4. The authors use -4°C for cold stress but it should be freezing stress according to figure S6. Please clarify.
5. What is the base to use the salt treatment as other stresses like heat, cold (4°C), and flood can be used for the expression analysis according to the figure S6 data?
6. Is the legend for fig 8A and B the same? Please check.
Author Response
Response to the Reviewer
Thank you for giving us the opportunity to submit a revised draft of our manuscript. We appreciate the time and effort that you dedicated to providing feedback on our manuscript and are grateful for the insightful comments and valuable improvements to our paper. Please see below, in blue, for a point-by-point response. We have also highlighted the changes within the manuscript.
Comment 1; The manuscript “Functional characterization of the cystine-rich-receptor-like ki- 2 nases (CRKs) and their expression response to Sclerotinia sclerotiorum in Brassica napus” is well structured and the aim of the research work is relevant. I have the following concerns regarding the current manuscript that authors should consider while revising their manuscript.
The authors should abbreviate the genus name after the first use in the text. Like Sclerotinia sclerotiorum, Solanum lycopersicum, and Oryza sativa. Please check throughout the manuscript.
Response; Thank you for pointing this out, as per the reviewer suggestion, we have abbreviated the Sclerotinia sclerotiorum, Solanum lycopersicum, and Oryza sativa as S. sclerotiorum, S. lycopersicum, and O. sativa, respectively in the revised manuscript.
Comment 2; Statistical analysis should be included in "Materials & Methods". In what software the statistical analysis was carried out? How Statistical Significance was determined? What statistical test was used?
Response; The Graphpad program version 8.0 was used to perform the statistical significance analysis and the One-way ANOVA followed by Dunnett’s t-test was utilized to find the significant difference between all stress treatments with the controls. The p-value below 0.005 was determined as statistically significant and indicated with the asterisk sign. As per the reviewer suggestion, we have added this paragraph in the material and method section lines 558-561 please check in the revised manuscript.
Comment 3; In Figure 9 graphs (x-axis), Sclerotinia sclerotiorum should be italic.
Response; Thank you for pointing this out, we have corrected it.
Comment 4; The authors use -4°C for cold stress but it should be freezing stress according to figure S6. Please clarify.
Response; Thank you for pointing this out, we are very sorry about our typo error, it is freezing stress, we have corrected it in the revised manuscript and also in the figure 9.
Comment 5; What is the base to use the salt treatment as other stresses like heat, cold (4°C), and flood can be used for the expression analysis according to the figure S6 data?
Response; Most of the previous studies have reported the role of cystine rich receptor like kinases in the plant defense against several stresses, including pathogen infection, oxidative stress, and abiotic stresses [1-5] however, only a few studies have shown the response of CRK under salt treatment [6], except for heat and flood. Additionally, the salt content is considered one of the main abiotic factors affecting the yield of B. napus. Which prompted us to ask whether B. napus CRKs play a role in salinity tolerance. Therefore, to fully characterized and understand the expression profile of the BnaCRKs we observed their expression pattern in response to salinity and other treatments. However, we were unable to extract the significant data of BnaCRKs under heat and flood treatment from qPCR-analysis, thus we decided not to include it in this study. Furthermore, to overcome the reviewer's concern, we have extracted the expression data of BnaCRKs under 24 hours of salinity treatment from the RNA-seq dataset (Genome sequence archive accession: CRX052409) and added in figure S6 and Table S9.1 please check in the revised manuscript.
Comment 6; Is the legend for fig 8A and B the same? Please check.
Response; Thank you for pointing this out, it’s not the same, fig 8A shows the distribution of Light responsive, stress and defense-related, hormone-responsive, and development-related cis-acting elements in the promoter region of BnaCRKs, whereas fig 8B represents the total number of each cis-element. We have corrected it in the revised manuscript please check the figure 8 legend.
References
- Baba, A. I.; Rigó, G.; Ayaydin, F.; Rehman, A. U.; Andrási, N.; Zsigmond, L.; Valkai, I.; Urbancsok, J.; Vass, I.; Pasternak, T., Functional Analysis of the Arabidopsis thaliana CDPK-Related Kinase Family: At CRK1 Regulates Responses to Continuous Light. International journal of molecular sciences 2018, 19, (5), 1282,
- Yadeta, K. A.; Elmore, J. M.; Creer, A. Y.; Feng, B.; Franco, J. Y.; Rufian, J. S.; He, P.; Phinney, B.; Coaker, G., A cysteine-rich protein kinase associates with a membrane immune complex and the cysteine residues are required for cell death. Plant Physiology 2017, 173, (1), 771-787,
- Li, T.-G.; Zhang, D.-D.; Zhou, L.; Kong, Z.-Q.; Hussaini, A. S.; Wang, D.; Li, J.-J.; Short, D. P.; Dhar, N.; Klosterman, S. J., Genome-wide identification and functional analyses of the CRK gene family in cotton reveals GbCRK18 confers verticillium wilt resistance in Gossypium barbadense. Frontiers in plant science 2018, 9, 1266,
- Zhang, X.; Yang, G.; Shi, R.; Han, X.; Qi, L.; Wang, R.; Xiong, L.; Li, G., Arabidopsis cysteine-rich receptor-like kinase 45 functions in the responses to abscisic acid and abiotic stresses. Plant Physiology and Biochemistry 2013, 67, 189-198,
- Lu, K.; Liang, S.; Wu, Z.; Bi, C.; Yu, Y.-T.; Wang, X.-F.; Zhang, D.-P., Overexpression of an Arabidopsis cysteine-rich receptor-like protein kinase, CRK5, enhances abscisic acid sensitivity and confers drought tolerance. Journal of Experimental Botany 2016, 67, (17), 5009-5027,
- Bourdais, G.; Burdiak, P.; Gauthier, A.; Nitsch, L.; Salojärvi, J.; Rayapuram, C.; Idänheimo, N.; Hunter, K.; Kimura, S.; Merilo, E., Large-scale phenomics identifies primary and fine-tuning roles for CRKs in responses related to oxidative stress. PLoS Genetics 2015, 11, (7), e1005373,
